# Remdesivir Treatment for COVID 19 in Pregnant Patients with Moderate to Severe Symptoms: Serial Case Report

**Yudianto Budi Saroyo** [1]**, Amanda Rumondang** [2,*]**, Irene Sinta Febriana** [2]**, Achmad Kemal Harzif** [3]
**and Rima Irwinda** [1]

1   Obstetric Gynecology Department, Maternal Fetal Medicine Division, Cipto Mangunkusumo
    General Hospital, Jakarta 10430, Indonesia; yudibs@gmail.com (Y.B.S.); rima.irwinda@yahoo.com (R.I.)
2   Obstetric Gynecology Department, Cipto Mangunkusumo General Hospital, Jakarta 10430, Indonesia;
    irene.sintafebriana@gmail.com
3   Reproductive Endocrinology and Infertility Division, Obstetric Gynecology Department,
    Cipto Mangunkusumo General Hospital, Jakarta 10430, Indonesia; kemal.achmad@gmail.com
*   Correspondence: amanda17.md@gmail.com

**Abstract:** Introduction: Severe Acute Respiratory Syndrome Corona Virus 2 (SARS-CoV-2) infection that causes *novel* Coronavirus Disease 2019 (COVID-19) has become a major health problem worldwide and been declared a pandemic since March 2020 by WHO. One special population that poses a challenge is pregnant women with COVID-19. There have not been many studies related to COVID-19 in pregnancy. In this study, we present five serial cases of Remdesivir treatment for COVID-19 in pregnant women with moderate to severe symptoms. Case Illustration: We briefly describe five serial cases being treated with Remdesivir therapy during hospitalization. Four cases were delivered by cesarean section, and one was delivered vaginally in gestation week 37. All cases showed a shortened duration of hospitalization, rapid improvement in clinical symptoms, and no adverse events were observed in mothers, fetuses, and neonates. Discussion: Remdesivir, an inhibitor RNA Polymerase, has been used in COVID-19 treatment and is known to shorten recovery time in nonpregnant women. Some studies have shown no adverse effects on Remdesivir for pregnant women. Based on randomized control trial (RCT) during the Ebola epidemic, Remdesivir was safe to use for pregnant women. All cases showed reduced hospitalization time and better clinical outcomes without maternal, fetal, or neonatal adverse events. Conclusion: Remdesivir protocol for pregnant women with moderate to severe symptoms of COVID-19 has resulted in better clinical improvement with a shorter recovery period and no adverse effects during the hospitalization period. Further studies and RCT are warranted to evaluate the biosafety and effects of Remdesivir in pregnant women.

**Keywords:** Remdesivir; pregnancy; COVID-19; moderate–severe symptoms

## 1. Introduction

Severe Acute Respiratory Syndrome Corona Virus 2 (SARS-CoV-2) infection that has caused the novel Coronavirus Disease 2019 (COVID-19) has become a major health problem worldwide with thousands of mortality cases. On 11 March 2020, World Health Organization (WHO) declared COVID-19 as a pandemic [1,2]. Clinical symptoms of this disease are variable, ranging from asymptomatic to hypoxemic respiratory failure and even death. On 19 January 2021, WHO reported more than 4.7 million new cases for the past week, with the total cumulative cases over 93 million and over 2 million deaths worldwide since the beginning of the pandemic [3].

Physiological and immune system changes during pregnancy lead to increased sus-ceptibilities toward viral infection, hence worsening its outcome [1,4]. The US Center for Disease Control and Prevention (CDC) announced that there were 11.764 cases of COVID 19 in pregnancy, from 29 March 2020 until 10 February 2021. Most cases contracted the

infection during the second trimester (2337 cases) [5]. Allotey et al. reported that 10% of pregnant women admitted to the hospital were diagnosed as suspected or confirmed COVID-19 cases (11,432 cases) [6].

COVID-19 has a spectrum of clinical manifestations, ranging from mild to severe/critical [7]. In most pregnant women, the clinical symptoms appear to be mild, similar to the nonpregnant adult population, while approximately 14% of pregnant women infected by SARS-CoV-2 have severe or critical symptoms [6,8,9]. Mild symptoms of COVID 19 manifest as fever, cough, anosmia, and no dyspnea, while moderate symptoms manifest as lower respiratory tract disease, with oxygen saturation >94% and radiographic evidence. In severe symptoms, oxygen saturation decreases <94% with a respiratory rate of >30×/min and radiographic findings of lung infiltrates >50%. Patients with respiratory failure, multiorgan dysfunction, or even failure are considered critical [7]. The appearance of viral pneumonia has become the common cause of death in pregnant women worldwide [1].

Diagnosis of COVID-19 in pregnant patients is performed following the same protocols as those in the nonpregnant population. Confirmed COVID-19 cases are diagnosed using a Real Time-Polymerase Chain Reaction (RT-PCR) examination from a naso-oropharyngeal swab. In some clinical conditions with moderate to severe dyspnea, Computerized Tomography (CT) imaging of thorax is performed to add diagnostic value and predict patient prognoses [1,9].

As a special population, managing COVID-19 in pregnant women presents a challenge [1,4,8]. There is still no standardized treatment in pregnant women with COVID-19, both at national and international levels. Pregnant women were often excluded from some studies to evaluate novel therapy for COVID-19. However, these data are needed to determine the efficacy and safety of COVID-19 management in pregnancy [1,2].

Remdesivir, a viral RNA polymerase inhibitor, has been used to manage severe COVID-19 in nonpregnant adults. Randomized controlled trial during Ebola epidemic has shown that Remdesivir can be safely used in pregnancy. However, data regarding the use of Remdesivir in pregnant women with moderate to severe COVID-19 are limited.

Remdesivir included in our hospital guidelines for COVID-19 treatment as a choice of antiviral to be given to pregnant women with symptomatic COVID-19. This study report has successfully managed five cases of COVID-19 in pregnancy with moderate to severe SARS-CoV-2 infection using Remdesivir during their hospitalization in Dr. Cipto Mangunkusumo National Central General Hospital.

## 2. Cases

### 2.1. Case A

Mrs. K, a 27-year-old pregnant woman at $27^{+4}$ weeks of gestation, presented to our obstetric emergency unit with tachypnea (RR 28×/min; saturation 92%) one day before admission. The patient also had a fever (103.1 °F) and cough 1 week before admission. Other than that, she also had chronic hypertension. A chest X-ray showed patchy consolidation that corresponded to bilateral pneumonia. A naso-oropharyngeal swab for SARS-CoV-2 was positive as tested by RT-PCR (RdRP Cq 35.15; E Gene Cq 32.10). The patient was taken care of in the High Care Unit (HCU) with oxygen therapy using 10 L per minute (Lpm) with a nonrebreathing mask (NRM), and $O_2$ saturation maintained at 95–97%. She was diagnosed with acute respiratory distress and superimposed preeclampsia with severe features. The patient was treated with anti-hypertension (nifedipine 4 × 10 mg), MgSO4 for seizure prevention, and dexamethasone 2 × 6 mg for lung maturation for 2 days.

On hospital care day 2, tachypnea worsened (RR 32×/min), with a decreased $O_2$ saturation to 90% while on oxygen therapy given at 10 Lpm with NRM. The patient had an emergency cesarean section, and a baby girl was born with a birthweight of 1150 g, birth length of 34 cm, and APGAR score (AS) 7/9. The baby was taken care of in the NICU isolation for COVID-19 as a naso-oropharyngeal swab for SARS-CoV-2 result was negative.

For postoperative observation, the patient was taken to HCU with no clinical improvement. On hospital care day 7 (HD 7), she was given Remdesivir protocol for 5 days.

Tachypnea and saturation improved on HD 8 (RR 24×/min; O$_2$ saturation 96% with NRM 6 Lpm). Both patient and baby were discharged on HD 12 after the SARS-CoV-2 swab came back negative.

### 2.2. Case B

Mrs. M, a 28-year-old pregnant woman at 36 weeks gestation, was diagnosed with confirmed COVID-19 and was referred to our hospital due to worsening clinical symptoms. The patient had been treated in a previous hospital for 2 days with symptoms of cough and fever 4 days before admission. During hospitalization, the patient's condition worsened as she displayed tachypnea (RR 32×/min, O$_2$ saturation 92%) and fever (102.2 °F).

A chest X-ray showed no abnormalities in both lungs. A naso-oropharyngeal swab for SARS-CoV-2 was positive (N gene Cq 39.54). Oxygen saturation was 95% using oxygen therapy at 3 Lpm with a nasal cannula. An emergency cesarean section was directly performed due to the worsening of the maternal condition and oligohydramnios seen from ultrasound examination. The cesarean section was performed with no complications. A baby boy was born with a birthweight of 2345 g, a birth length of 45 cm, and AS 7/8. The baby was taken to perinatal isolation for COVID-19 with a negative result for SARS-CoV-2 from a naso-oropharyngeal swab.

The patient was taken care of in HCU after surgery on day 1 postoperative (HD 1). Remdesivir was directly given with a dose of 200 mg IV on the 1st day and continued with 1 × 100 mg IV on day 2 until 5. The patient's postoperative condition was stable, and the RT-PCR result for SARS-CoV-2 was negative on HD 4. Both patient and baby were discharged on HD 7 after Remdesivir therapy was completed.

### 2.3. Case C

Mrs. N, a 31-year-old pregnant woman at 40 weeks gestation, presented to our obstetric emergency unit with tachypnea (RR 30×/min; O$_2$ saturation 93%) and fever (101.3 °F) for one week before admission. The patient had a history of flu-like symptoms and anosmia for the previous 2 weeks. A chest X-ray showed no appearance of pneumonia. Oxygen therapy was given with a nasal cannula, 4 Lpm, and O$_2$ saturation was 96%. SARS-CoV-2 (RdRP Cq 24.40; E Gene Cq 23.89) was confirmed using RT-PCR from a naso-oropharyngeal swab.

We immediately conducted a cesarean section due to a premature rupture of the membrane with oligohydramnios. The cesarean section was successful, and a born baby boy was born, 3465 g, 51 cm, AS 8/9. The neonate was taken care of in the perinatal isolation unit for COVID-19, and a naso-oropharyngeal swab for SARS-CoV-2 result came back negative.

The patient was taken to the isolation ward for COVID-19 for postoperative observation. The patient was given Remdesivir on HD 2 until HD 6. The patient's symptoms improved during hospitalization with a negative swab result for SARS-CoV-2 on hospital care day 6. Both patient and baby were discharged with a stable condition on HD 7.

### 2.4. Case D

Mrs. T, a 30-year-old at 32 weeks gestation with two histories of cesarean section, presented to our obstetric emergency unit with tachypnea (RR 30×/min; O$_2$ saturation 93%), cough, and fever (102.2 °F). The patient had complained of tachypnea for 7 days before admission. Her naso-oropharyngeal swab was positive for SARS-CoV-2 by real-time PCR (RdRP Cq 26.09; E Gene Cq 26.10) without abnormalities in a chest X-ray.

The patient was treated in HCU and was given oxygen therapy 4 Lpm with a nasal cannula, and O$_2$ saturation was maintained at 95%. Remdesivir was given on HD 1 and continued for 6 days (HD 6). During hospitalization, the fetal condition was observed using cardiotocography, and no abnormalities were seen. Clinical symptoms improved, and the patient was discharged on HD 7 with a negative result from a naso-oropharyngeal swab for SARS-CoV-2. The patient underwent a cesarean section at 39 weeks gestation due

to a previous cesarean section. A baby girl was born, 2700 g, 49 cm, A/S 9/10. The baby was rooming with her mother after delivery.

*2.5. Case E*

Mrs. A, a 35-year-old at 27 weeks gestation, was referred to our institution with a fever (101.3 °F), flu-like syndrome, tachypnea (RR 28×/min; $O_2$ saturation 94% room air), and anosmia. The patient had complained of symptoms for 4 days before admission. A chest X-ray was normal. A naso-oropharyngeal swab was positive for SARS-CoV-2 by real-time PCR (RdRP Cq 15.21; E Gene Cq 15.58) without abnormalities in a chest X-ray.

The patient was taken care of in an isolation ward for COVID-19 and given oxygen therapy 4 Lpm with a nasal cannula (saturation 96–97%). Remdesivir was given on HD 1 until HD 6. Daily fetal monitoring was performed using a fetal doppler and fetal kick count chart. During hospitalization, clinical symptoms improved. The patient was discharged after 7 days of hospitalization with a repeated naso-oropharyngeal swab for SARS-CoV-2 result which came back negative. The patient underwent vaginal delivery at 37 weeks gestation. A baby boy was born, 3100 g, 51 cm, A/S 8/9. Both mother and baby were in good condition after delivery.

Remdesivir Protocol:

Day 1: 200 mg IV single dose

Day 2–5: 100 mg IV/daily

## 3. Discussion

Remdesivir, an inhibitor RNA Polymerase, has been used in COVID-19 treatment and is known to shorten recovery time in nonpregnant women [8,10,11]. Unfortunately, pregnant women were being excluded in most studies. Some studies have shown that Remdesivir resulted in no adverse effects for pregnant women. Based on a randomized control trial (RCT) performed during the Ebola epidemic, Remdesivir was safe to use for pregnant women [2,11].

In our study, no adverse events were observed from the Remdesivir therapy for both mothers and fetuses (Case A, C, and E). There was delayed Remdesivir treatment on case A (given on HD 7). However, after Remdesivir administration, the clinical condition rapidly improved during hospitalization. Our cases also showed that the period of hospitalization was shorter. This result was not only supported by better clinical outcomes but also with an improvement in laboratory findings and negative results of SARS-CoV-2 from naso-oropharyngeal swabs shortly after finishing the therapy. Most patients were taken care of for 7 days (case B–E), with 6 days of Remdesivir protocol. During therapy administration, no unwanted events took place for both mothers and fetuses.

Some studies have shown the safety effects of Remdesivir in pregnancy [2,8,11]. The FDA has recently granted emergency use of Remdesivir for severe COVID-19 in vulnerable populations, including pregnant women. A study conducted by Pierce showed that 16% of pregnant women with severe symptoms of COVID-19 treated with Remdesivir had better clinical improvement during hospitalization [8,12]. However, more data are still needed to conclude the safety and effectiveness of the Remdesivir protocol for pregnant women infected by COVID-19. RCTs to investigate the effectiveness of Remdesivir in treating COVID-19 in pregnancy are still needed. RCT studies are still unable to be conducted in our institution for pregnant women with moderate to severe COVID-19 symptoms taking ethical issues into consideration.

Other antivirals that can be considered for pregnant women with COVID-19 are Oseltamivir and Lopinavir/Ritonavir [13,14]. Oseltamivir, a neuraminidase inhibitor, has been used during the Novel Influenza A (H1N1) pandemic in 2009. Based on previous studies, there were some adverse effects on fetuses, especially if it was given in the first trimester. Adverse effects due to Oseltamivir in the first trimester are spontaneous abortion (6.1%), therapeutic abortions (11.3%), and nine cases of birth defects (Ventricular Septal

Defect, Anophthalmos). Besides that, the effectiveness in reducing symptoms is lower as compared to Remdesivir, particularly in moderate to severe COVID-19 symptoms [13].

Lopinavir-Ritonavir, a component of HAART treatment, is frequently used for the treatment of HIV in pregnancy. This antiviral is used in order to prevent mother-to-child transmission. The efficacy in treating COVID-19 in pregnancy remains unclear and warrants further studies [14].

In our observations, there were increasing D-Dimer levels in our patients (A–E) (Table 1). Low-molecular-weight heparin (LMWH) treatment was given to patients D and E, while patients B and C did not get heparin therapy as a cesarean section was immediately performed. In postoperative observations, the D-Dimer level was decreasing (See supplementary material). Patient A also did not get LMW as clinical improvements were observed postoperatively, but no complete data could be obtained in regard to the patient's D-Dimer levels (Table 1). LMWH is known to decrease the risk of thromboembolic in pregnant patients with COVID 19 and also to improve prognosis with severe symptoms [15]. During Remdesivir and LMWH therapy, we observed no adverse events and worsening of conditions in patients, fetuses, and also neonates. But further studies are still needed to evaluate the adverse events from the interactions of these two medications.

**Table 1.** Laboratory findings in pregnant women treated with Remdesivir protocol.

| Laboratory Component | | A | | B | | C | D | | E | |
|---|---|---|---|---|---|---|---|---|---|---|
| Reference Range and Units | Reference Range and Units | Rmdsvr D1 | Rmdsvr D5 | Rmdsvr D1 | Rmdsvr D6 | Rmdsvr D1 | Rmdsvr D1 | Rmdsvr D4 | Rmdsvr D1 | Rmdsvr D6 |
| Hemoglobin (g/dL) | 11.7–15.7 | 12.9 | 12.7 | 13.6 | 12.7 | 12.0 | 10.3 | 10.1 | 10.5 | 10.1 |
| Erythrocytes ($10^6$/uL) | 3.80–4.80 | 4.33 | 4.34 | 4.25 | 3.97 | 4.63 | 3.95 | 3.81 | 3.65 | 3.53 |
| WBC Count ($10^3$/uL) | 150–410 | 14.21 | 18.92 | 14.18 | 19.68 | 5.31 | 8.86 | 12.83 | 8.25 | 6.14 |
| Thrombocytes ($10^3$/uL) | 4.0–11.0 | 298 | 646 | 344 | 360 | 307 | 265 | 355 | 270 | 226 |
| MCV/VER (fL) | 83.0–101.0 | 88 | 86.9 | 88.7 | 89.9 | 79.3 | 79.5 | 79.5 | 84.1 | 83.0 |
| MCH/HER (pg) | 27.0–32.0 | 29.8 | 29.3 | 32.0 | 32.0 | 25.9 | 26.1 | 26.5 | 28.8 | 28.6 |
| MCHC/KHER (g/dL) | 31.5–34.5 | 33.9 | 33.7 | 36.1 | 35.6 | 32.7 | 32.8 | 33.3 | 34.2 | 34.5 |
| Basophil (%) | 0–2 | 0.5 | 0.1 | 0.8 | 0.3 | 0.4 | 0.2 | NA | 0.4 | 0.2 |
| Eosinophil (%) | 1–6 | 0.1 | 0.1 | 0.0 | 0.0 | 1.5 | 0.1 | NA | 0.0 | 0.2 |
| Neutrophil (%) | 40.0–80.0 | 78.1 | 91.2 | 78.1 | 84.9 | 69.4 | 72.7 | NA | 73.3 | 70.8 |
| Lymphocyte (%) | 20–40 | 14.2 | 6.3 | 15.2 | 10.6 | 20.0 | 13.2 | NA | 13.7 | 21.3 |
| Monocyte (%) | 2–10 | 0.0 | 2.3 | 5.9 | 4.2 | 8.7 | 13.8 | NA | 12.6 | 7.5 |
| Neutrophiil Count ($10^3$/uL) | 1.70–7.50 | 11.10 | 17.26 | 11.09 | 16.73 | 3.69 | 6.44 | NA | 6.05 | 4.35 |
| Lymphocyte Count ($10^3$/uL) | 1.00–3.20 | 2.02 | 1.20 | 2.15 | 2.08 | 1.06 | 1.17 | NA | 1.13 | 1.31 |
| NLCR | None | 5.50 | 14.28 | 5.16 | 8.04 | 3.48 | 5.50 | NA | 5.35 | 3.32 |
| CRP (mg/dL) | <0.50 | 58.1 | NA | 41.6 | NA | 10.7 | 12.3 | 8.62 (Hs) | 17.9 | NA |
| Albumin (g/dL) | 3.50–5.20 | 3.71 | NA | NA | NA | NA | NA | NA | NA | NA |
| Creatinine (mg/dL) | 0.55–1.02 | 0.50 | 0.50 | 0.80 | NA | 0.90 | 0.60 | NA | NA | NA |
| AST (U/L) | 10–35 | 23 | 21 | 36 | NA | 30 | 25 | NA | NA | NA |
| ALT (U/L) | 0–55 | 23 | 57 | 23 | NA | 25 | 24 | NA | NA | NA |
| D-dimer (ug/mL) | <440 | NA | 3510 | 930 | NA | 2600 | 1500 | 3120 | 2020 | 1250 |
| Ferritin (ng/mL) | 20.0–200.0 | NA | 158.84 | NA | NA | NA | NA | NA | NA | NA |
| LDH (U/L) | 135–214 | NA | 287 | NA | NA | NA | NA | NA | NA | NA |
| Procalcitonin (ng/mL) | <0.05 | 0.06 | NA | 0.53 | NA | 0.08 | 0.06 | NA | NA | NA |
| Ureum (mg/dL) | 15–40 | 14.7 | 35.4 | 19.2 | NA | 23 | 16.0 | NA | NA | NA |

Our experience in managing pregnant women with COVID-19 using the Remdesivir protocol was uneventful. Some limitations in this study were short period in which the study was conducted, and there were only a few cases. Another limitation is that no control case was used as a comparison to evaluate the effectiveness of Remdesivir.

Although our study has some limitations, we strongly suggest the use of the Remdesivir protocol for pregnant women with moderate to severe symptoms of COVID-19, pend-

ing FDA approval, remembering the current time of pandemic we are in that necessitates emergency protocol and also no observation of harmful events.

There is still no standardized therapy, including an antiviral regiment, that is recommended for treatment of COVID-19 in pregnancy, especially in moderate–severe cases. A randomized controlled trial is also unethical to perform in these cases. Further studies with more cases are needed with the inclusion of pregnant women in Remdesivir clinical trials to give a perspective on its safety, efficacy, and adverse effects towards the mothers, fetuses, and neonates.

## 4. Conclusions

Remdesivir protocol for pregnant women with moderate to severe symptoms of COVID-19 showed no adverse effects during hospitalization, clinically improved, and shortened recovery during hospitalization. Further study and randomized controlled trial (RCT) are needed to evaluate biosafety and its effect on the Remdesivir protocol in pregnant women.

**Supplementary Materials:** All supplementary materials are available online at https://www.mdpi.com/article/10.3390/idr13020042/s1, Table S1: Laboratory findings in pregnant women treated with Remdesivir protocol.

**Author Contributions:** Y.B.S.: Designing idea and revising manuscript; A.R.: Collecting data, writing manuscript and revising manuscript; I.S.F.: Collecting data and writing manuscript; A.K.H.: Designing idea, collecting data and revising manuscript; R.I.: Writing manuscript and revising manuscript. All authors have read and agreed to the published version of the manuscript.

**Funding:** This study received no external funding.

**Institutional Review Board Statement:** Not applicable. Medication therapy during this study was included in our hospital guidelines for COVID-19 treatment for choice of antiviral to be given to pregnant women with symptomatic COVID-19 infection. Our recent Panduan Praktek Klinik (PPK) (Institutional guidelines) for COVID-19 Treatment was renewed in 2021.

**Informed Consent Statement:** Informed consent was including in general consent for hospitalized patient. No special informed consent form since this treatment was including in our institutional guidelines for COVID-19 treatment.

**Data Availability Statement:** All data completely presents in paper.

**Acknowledgments:** Thank you to all colleagues, midwives, and staff in the Obstetric Gynecology Department, Cipto Mangunkusumo General Hospital, Faculty of Medicine Universitas Indonesia, who supported the gathering and compiling this serial case.

**Conflicts of Interest:** The authors declare no conflict of interest.

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
