# Peer review of "Remdesivir Treatment for COVID 19 in Pregnant Patients with Moderate to Severe Symptoms: Serial Case Report"

_2036-7449, doi:10.3390/idr13020042_

Round 1

Reviewer 1 Report

I selected minor revision because the publication is worth publishing.The weakness of work is related to the low number of respondents, however, it is related to the determinants of the problem and the short period of possibility of recruiting patients.
I would include this publication in the case report (multicase report?) 1. The introduction should short describe standard procedures, both diagnostic and therapeutic, in the case of severe infections. It should be specified in which cases retroviral therapy should be initiated. 2. In the discussion, I miss information on alternative reatments to remdesivir and whether other anti-viral drugs have been studied in pregnancy. If not for humans, then for example in animal models.

Thank you very much for the opportunity to review this very interesting publication. During a pandemic, publishing such research can be very useful in certain difficult cases. Thanks to them, we can resist the use of innovative therapies and legalize them. In the introduction, I propose to read the publication DOI: 10.3390/jcm9113749. I hope that the publication will be available soon and will help doctors in making the optimal therapeutic decisions.

Author Response

Thank very much for reviewing the manuscript. I do agree and already adding the information regarding your review :

  1. Standard diagnostic and treatment
  2. Alternative anti virus other than remdesivir

Hereby i attach the revise manuscript in green highlight.

Reviewer 2 Report

1. Please clarify "moderate-severe symptoms" in the manuscript including
title. Do you mean moderate to severe?

2. RCTs are the gold standard to evaluate the effectiveness of
medications. Please discuss the limitations of the present study design
in more detail. How can the authors assure that “beneficial effects” are
due to the medication and not simply due to natural improvements over
time or improvements by chance?

3. The authors may want to discuss how to advance this research in the
future. What are the urgent needs? How can they be achieved given the
time pressure in this crisis?

4. Please check English language very carefully.

Author Response

Thank you very much for reviewing the manuscript. Hereby i attach the revise version regarding :

  1. Clarification on moderate to severe symptoms
  2. The RCTs as gold standard including limitations
  3. The urgency to advance this study during this pandemic condition

Thank you in advance.

Round 2

Reviewer 2 Report

I don't have further questions.